# Nanotechnology-Based Diagnostics for Diseases Prevalent in Developing Countries: Current Advances in Point-of-Care Tests

**DOI:** 10.3390/nano13071247

**Published:** 2023-03-31

**Authors:** Lungile Nomcebo Thwala, Sphumelele Colin Ndlovu, Kelvin Tafadzwa Mpofu, Masixole Yvonne Lugongolo, Patience Mthunzi-Kufa

**Affiliations:** 1National Laser Centre, Council for Scientific and Industrial Research, P.O. Box 395, Pretoria 0001, South Africa; 2College of Agriculture, Engineering and Science, School of Chemistry and Physics, University of Kwa-Zulu Natal, University Road, Westville, Durban 3630, South Africa

**Keywords:** nanotechnology, nanoparticles, diagnostics, infectious, non-infectious, point of care

## Abstract

The introduction of point-of-care testing (POCT) has revolutionized medical testing by allowing for simple tests to be conducted near the patient’s care point, rather than being confined to a medical laboratory. This has been especially beneficial for developing countries with limited infrastructure, where testing often involves sending specimens off-site and waiting for hours or days for results. However, the development of POCT devices has been challenging, with simplicity, accuracy, and cost-effectiveness being key factors in making these tests feasible. Nanotechnology has played a crucial role in achieving this goal, by not only making the tests possible but also masking their complexity. In this article, recent developments in POCT devices that benefit from nanotechnology are discussed. Microfluidics and lab-on-a-chip technologies are highlighted as major drivers of point-of-care testing, particularly in infectious disease diagnosis. These technologies enable various bioassays to be used at the point of care. The article also addresses the challenges faced by these technological advances and interesting future trends. The benefits of point-of-care testing are significant, especially in developing countries where medical care is shifting towards prevention, early detection, and managing chronic conditions. Infectious disease tests at the point of care in low-income countries can lead to prompt treatment, preventing infections from spreading.

## 1. Introduction

Identifying a disease and its underlying causes is crucial for managing and treating the condition. This involves obtaining information about the patient’s medical history and physical examination, as well as using diagnostic techniques to detect and describe signs and symptoms. Ideally, diagnostic tools should be sensitive, accurate, specific, robust, user-friendly, and cost-effective. Although traditional diagnostic methods have played a significant role in disease diagnosis, they have limitations, such as being slow, expensive, inaccurate, and requiring trained professionals, particularly in developing nations. As a result, new and innovative diagnostic techniques, such as nanodiagnostics, are necessary to meet the growing demand for early detection, high sensitivity, and point-of-care testing, especially for community clinics in developing countries. Developing countries face greater exposure and are more vulnerable to various challenges stemming from multiple factors, such as geography, demographics, and socio-economic conditions. Diseases are hindering human progress in these countries, with adverse effects on education, income, life expectancy, and other health metrics.

Nanodiagnostics is an emerging field that utilizes nanoscale properties to manipulate and analyze single-molecule systems and platforms for clinical diagnostics [1]. With their unique physiochemical and optical characteristics, nanomaterials have greatly contributed to accurate and timely disease diagnosis [2]. As many diseases can lead to epidemics and high morbidity and mortality rates, it is becoming increasingly crucial to develop nanotechnology-based diagnostic methods that are fast, vigorous, user-friendly, and cost-effective for detecting clinical samples. 

In recent years, a significant number of nanomaterials for biomedical applications have been developed and reported. In 2022, Dukle et al. reported advances on the development of paper-based sensors that can be applicable in cancer screening, especially in low-income countries. The review highlighted the types of paper-based sensors, the analytes that can be detected, and the detection and readout methods, with an emphasis on the need for low-cost, portable devices that should not require any special training [3]. During the same year, Yang et al. detailed microfluidic POC equipment and its applications in detecting infectious diseases, cardiovascular diseases, cancer, and chronic diseases. The future incorporation of microfluidic POC devices into applications such as wearable devices and telemedicine was also discussed. This review also called attention to the status of disease treatment in underdeveloped areas [4]. Chinnadayyala et al. reviewed various Electrochemiluminescence (ECL) signal amplification labels for inexpensive and portable devices, which have replaced traditional instrumentation and different light detection technologies used in paper ECL devices. They also highlighted the current trends and developments in ECL paper-based microfluidic analytical devices, as well as recent applications of ECL-based detection methods and inexpensive microfluidic devices. Significant efforts have also been dedicated towards how miniaturized systems, using immobilized reagents on cellulose paper and nanoscale technology, have enhanced the analytical performance of these ECL biosensors with implications for point-to-care diagnostics [5]. Considering these progressive developments, it is necessary to highlight the latest advancements related to the use of nanomaterials for enhancing the performance of POC devices in a cost-effective manner. Hence, the current review provides some recent advances in the development of POC testing devices where nanosystems have been used to enhance their function. The review focuses on devices that have a great potential to be applicable in low-income, developing countries which are mainly affected by low survival rates when it comes to diseases of socio-economic importance. This review begins with an overview of selected infectious diseases prevalent in developing countries and their conventional testing methods are outlined. Thereafter, POC testing devices that have benefited from nanotechnology and are potentially applicable in developing countries are highlighted. Then, the writers report on advances in POC testing devices for detecting non-infectious chronic diseases, focusing on diabetes as the selected example. Finally, future trends as well as challenges that stand in the way of further progress on cost-effective POC testing devices are summarized.

## 2. Infectious Diseases

Pathogenic microorganisms, such as viruses, fungi, bacteria, and parasites, are the main culprits behind infectious diseases. Their unique characteristics, including rapid multiplication, unpredictability, and evolutionary advantage, have a profound impact on humanity [6]. Infectious diseases are responsible for approximately 15 million deaths worldwide, with respiratory infections and HIV being the leading causes [7]. Viral infections alone present significant global health challenges by affecting millions of individuals, leading to adverse health outcomes and hindering socio-economic development. These illnesses are transmissible and can quickly spread in certain areas, resulting in epidemics. Swift and accurate diagnosis of initial infections is crucial in preventing further spread, which can be achieved through in vitro diagnosis by detecting disease- or pathogen-specific biomarkers [8].

### 2.1. Viral Infections

Viruses, the smallest microbes, can only replicate inside living cells of an organism. They consist of a genetic core of DNA or RNA, enclosed in a protein capsid. Viruses need to bind to specific receptors on host cells in order to infect them. These receptors are often normal molecules that the cell needs for its own functions [9]. For instance, HIV-1 uses the CD4 receptor and chemokine receptors called CCR5 or CXCR4 [10]. Influenza viruses attach to sialic acid residues on cell surface glycoproteins, while SARS-CoV-2 requires the ACE2 receptor. Attachment is followed by viral entry into the cell, a process followed by replication. DNA viruses use the cell’s proteins and enzymes to copy their DNA, while RNA viruses use their RNA as a template for replication [11]. Viral nucleic material directs the host cell to synthesize viral enzymes and capsid proteins, resulting in the assembly of new virions [12]. The new virions are then released from the host organism, infecting adjacent cells and repeating the replication cycle.

Viral infections are typically diagnosed by detecting viral nucleic acid using PCR or LAMP, which amplifies DNA strands [13]. Unlike PCR, the LAMP method of amplification does not require complex equipment, as the reaction is held at a consistent temperature of 60–65 °C [12]. These molecular methods are the most specific and sensitive diagnostic tests. On the other hand, immunoassay-based methods detect viruses using antibodies produced in response to viral infections [14]. When someone contracts a virus, their body will begin producing antibodies in their bloodstream that are designed to target that specific virus. The presence of IgM in the blood indicates acute infection, while IgG indicates a past infection. 

### 2.2. Bacterial Infections 

Bacterial infections have a significant impact on public health, although they are generally easier to treat than viral infections. However, the problem of bacterial resistance to antimicrobials is rapidly growing and could have devastating consequences. Bacteria are prokaryotic organisms that store their genetic information in circular, double-stranded DNA molecules [15]. As an example, tuberculosis is the seventh most common cause of death globally and the main cause of death from a single infectious agent, ranking above HIV/AIDS and SARS-CoV-2. More than a quarter of tuberculosis deaths occur in the African region [16]. During infection, the first stage occurs within a week of inhaling the TB bacillus. Once the bacillus reaches the lung’s alveoli, special cells of the immune system called macrophages pick it up. The outcome of a tuberculosis infection is determined by the number of tuberculosis bacteria present and the ability of the macrophages to fight off the infection. When the number of bacteria is too high, or the macrophages are weakened, the bacteria can reproduce inside the macrophages, causing them to break down and allowing the bacteria to infect other nearby macrophages. If the macrophages cannot control the bacteria, the infection progresses to its second stage [17]. Laboratory testing for TB includes stained smears, culture, tuberculin skin test, Lipoarabinomannan (LAM) antigen detection, and molecular tests such as the Xpert MTB/RIF assay and Line probe assay [18].

### 2.3. Parasite-Caused Infections

Parasitic diseases are a widespread problem affecting millions of individuals worldwide, particularly in developing countries. Unfortunately, the treatment of these diseases is often limited. Malaria, for example, is caused by parasites that are transmitted to humans through the bites of infected female *Anopheles* mosquitoes. The initial inoculated sporozoite stage does not cause any symptoms and is transient. The parasite then infects liver cells and undergoes intracellular replication, with no symptoms. Once the liver replication stage is complete, the parasite enters the bloodstream and initiates the blood stage infection, which is the main cause of the disease symptoms. During this stage, merozoites break out of the liver and invade red blood cells, where they grow, divide, and destroy blood cells in the process.

Diagnosing malaria involves identifying the presence of malaria parasites or antigens in the patient’s blood. Various methods can be used, including microscopy, rapid diagnostic tests (RDTs), loop-mediated isothermal amplification (LAMP), and polymerase chain reaction (PCR). The choice of diagnostic tool may depend on several factors, such as the skills of laboratory personnel, patient load, and malaria epidemiology in the region. Microscopy is currently the gold standard for laboratory confirmation of malaria, where a drop of the patient’s blood is examined under a microscope after being spread as a ’blood smear’ on a slide and stained. PCR can detect parasite nucleic acids and may be more sensitive than smear microscopy, but it is not always practical in standard healthcare settings due to time constraints. PCR results may not be available quickly enough to confirm a diagnosis or establish the species of malaria parasite after diagnosis by smear microscopy or RDT [19,20].

## 3. Point-of-Care (POCT) Assays for Infectious Diseases

Point-of-care (POC) diagnosis is a modern approach to medical testing, performed in close proximity to the patient’s care. It aims to provide rapid, accurate, and real-time detection of medical conditions at the point of need [21]. POC diagnostics have gained increasing attention in recent years, particularly in clinical medicine where on-site detection is in high demand, especially in areas with limited resources. This has led to the development of various methods, devices, and biosensors that can fulfill the need for POC diagnostic tools.

Nanotechnology has been instrumental in advancing clinical applications, including tissue engineering, drug delivery, bioimaging, and diagnostics [22]. In particular, nanodiagnostics have gained significant interest in the field of infectious diseases due to their distinctive features such as rapid detection, enhanced sensitivity, and potential for point-of-care testing. Novel and effective nanodiagnostics for infectious diseases have been developed, offering the potential to create portable, robust, and affordable POC diagnostic platforms for detecting infectious diseases in developing countries [1]. 

To be effective, POC diagnostic technologies should be disposable, cost-effective, easy to use, and portable. They should be capable of analyzing small volumes of bodily fluids, such as blood, saliva, and urine. Cost is an important factor for global health applications [23] and efforts should be made to reduce costs by using minimal expensive reagents, inexpensive manufacturing methods for mass production, and ensuring quality control. Miniaturization is also crucial for developing portable POC diagnostic devices. Moreover, the environmental conditions of resource-limited settings, such as insufficient water, unreliable electricity, high temperatures, and humidity, should be considered for the clinical use of medical diagnostic devices.

### 3.1. Nanoparticles 

Nanomaterials and nanostructures possess unique characteristics that enable the development of nanodiagnostic platforms capable of detecting diseases rapidly and in real-time using only small volumes of patient samples. One significant advantage of nanoparticles is their high surface area to volume ratio, which allows for dense coverage of molecules. This property creates multiple binding sites for disease biomarkers, resulting in a multivalent effect that increases the specificity and sensitivity of bioassays (Figure 1) [24]. The applications of nanotechnology in diagnosis include the detection of extracellular biomarkers for diseases and in vivo imaging [25], as highlighted in Table 1. 

Various types of nanoparticle, including fluorescent, magnetic, and metallic nanoparticles, have been effectively employed for diagnosing infectious diseases. Fluorescent nanoparticles serve as sensitive and photostable probes that can label multiple biological targets. These nanoparticles are easily synthesized using polymers that promote fluorophore encapsulation, generating stable and versatile nanoparticles that perform better than organic dyes. Furthermore, functionalization of fluorescent nanoparticles is straightforward due to the presence of functional groups such as carboxylic acids, amines, and esters. Magnetic nanoparticles (MNPs) are used as contrast agents in magnetic resonance imaging (MRI). MNPs conjugated with antibodies have also been utilized for immunomagnetic separation of nuclei acids, proteins, and pathogens. The shape and magnetic properties of MNPs can be modulated by varying synthesis parameters such as polymer addition time and temperature, as well as the use of specific capping agents and surface modification, facilitating the addition of functional groups for attaching different ligands such as antibodies, proteins, and nucleic acids for target identification and quantification.

Conjugated polymer nanoparticles (CPNs) based on π-extended CPs exhibit high fluorescence brightness, low cytotoxicity, excellent photostability, reactive oxygen species (ROS) generation ability, and high photothermal conversion efficiency (PCE), endorsing them as an excellent theragnostic tool. CPNs-based probes for imaging, both in vitro and in vivo, have been explored [26,27,28]. By contrast, layered double hydroxides (LDHs) are widely researched as bioimaging agents because they are a suitable material in the development of nanocontrast agents due to their tunable properties based on their unique 2D crystal structure as well as their internal and external surface properties. LDH nanoparticle-based contrast agents are produced by doping various metal ions such as Gd3+, Mn2+, etc., through a suitable lattice manipulation within the positively charged metal hydroxide layer. The surface LDH nanoparticles can be functionalized by introducing photo-functional components, such as fluorescent probes or gold nanoparticles [29]. Hence, these nanoparticles have found vast applications both in disease diagnosis and therapy due to their high resolution and sensitivity.

Gold and silver nanoparticles are the most used metallic nanoparticles in diagnostics. They emit strong absorption when excited with electromagnetic radiation, and their surface chemistries can be altered by grafting or conjugating various probes such as antibodies and nucleic acids [30]. Gold and silver nanoparticles possess a surface plasmon band, typically at 422 nm for spherical-shaped nanoparticles, applicable in colorimetric detection and quantification of analytes. The plasmonic band shift depends on various factors such as particle size, chemical environment, adsorbed species, and dielectric constant. Changes in the local dielectric constant of nanoparticles caused by absorbed biomolecules or biomarker-induced agglomeration can cause plasmonic band shifts [31]. This unique feature of gold and silver nanoparticles enables the use of surface plasmon band shifts for various diagnostic applications by recording alterations in the UV-visible absorbance spectrum, as well as tags for target detection and quantification using fluorescence, Raman scattering, electrical conductivity, microscopy, and magnetic force techniques [30].

Quantum dots are nanostructures made of semiconductors that emit fluorescent light and have unique properties, including photostability, high quantum yield, and brightness. The size and composition of quantum dots can be varied to tune their fluorescence function and emission wavelength. These properties make them ideal for various biomedical applications, including detecting biomarkers such as antigens and pathogens [32,33]. As a successful example, Zhang P et al. used QD nanobeads as a signal amplification strategy in a dot-blot immunoassay [34]. On the other hand, carbon nanotubes (CNTs) have promising electrical conductivity and can be functionalized with numerous surface groups, such as hydroxyl or carboxyl groups, making them useful in biomedical applications [35,36,37]. CNTs are seamless nanotubes composed of carbon atoms, classified into single-walled nanotubes (SWCNTs) and multi-walled nanotubes (MWCNTs) founded on the number of layers of carbon atoms. Their large surface area allows for the loading of metal particles, biomolecules, or other polymorphs, improving their biocompatibility and amplifying detection signals. CNTs are primarily composed of sp2 hybridized carbon atoms, which confer excellent mechanical strength [37,38,39,40].

By utilizing nanosystems, it becomes possible to integrate multiple assays into a single device, resulting in reduced sample volumes, material consumption, and analysis time. These advantages contribute to the development of cost-effective, portable, and efficient devices, as discussed below. 

### 3.2. Microfluidic Devices

A microfluidic system/device is described as a small, non-turbulent, highly ordered, portable fluid flow system made of a set of microchannels molded or engraved into a material [4]. Much like electrical circuits, these microchannels can be engineered to perform complex tasks (their complexity can vary depending on the application). They are typically used in controlled biological experiments and their size is in the range of a few hundred micrometers. The microchannels forming the microfluidic chip are connected to achieve a set of desired features; for example, the channels can be connected so that the microfluidic system can perform sample pretreatment, separation, dilution, mixing, chemical reaction, detection, and product extraction (Figure 2) [4]. Microfluidic devices which integrate one or several of these laboratory functions on a single integrated circuit are referred to as lab-on-chip (LOC) devices [41]. Due to the range of applications in which they can be used, microfluidic systems can be thought of as a potential substitute for macroscale systems which are available in a typical bioscience and biomedical lab. Microfluidic diagnostic chips have special qualities that make them suitable for point-of-care applications, including modularity, mobility, low consumption of reagent and sample consumption, and high sensitivity [42].

Microfluidic systems have been shown to increase analysis speed and efficiency by automating the analysis process, reducing the consumption of samples or reagents (microfluidic systems typically work with microliter quantities of reagents) [43]. They can also be used to reduce human intervention in experiments [44], reduce experiment pollution [45], and allow for the efficient repeating of experiments through multiplexing [46]. Fluids in these microliter quantities have different behaviors compared to fluids on a macroscale; for example, they have low Reynolds or Grashof numbers. The Reynolds number is a dimensionless quantity which is used to predict flow patterns in different fluid flow situations. The Reynolds number measures the ratio between inertial and viscous forces. For low Reynolds numbers, we have laminar flow (smooth and streamlined flow), and we have turbulent flow (irregular and chaotic flow) for high/large Reynolds numbers [47].

The Reynolds number is given by Equation (1) below: (1)$$Re=\frac{\rho uL}{µ}.$$
where ρ is the density of the fluid (SI units: kg/m^3^), µ is the flow speed (m/s), *L* is the length of the channel where the fluid flows and µ is the dynamic viscosity of the fluid (N·s/m^2^). Typically, for *Re* < 2000, the flow is laminar fluid flow and becomes turbulent flow for *Re* > 4000. The difference between laminar and turbulent flow in molecular transport is the absence of convective and mixing flow in the laminar regime. 

Another quantity useful in microfluidics is the Péclet number, shown in Equation (2):(2)$$Pe=\frac{uL}{D}.$$
where *D* is the diffusion coefficient. The Péclet number [48] indicates the microfluidic mass transport, which is determined by the ratio of advective transport to diffusive transport along the channel. The Péclet number places convective and diffusive transport phenomena in correlation. Depending on the application of the biosensor, they can have a wide range of flow requirements. 

Reduced reagent use of microfluidics is beneficial in areas with scarce resources, such as disease detection at the point of care (POC), and when very small amounts of samples are available on demand. POC diagnostic testing devices are disease detection (testing) devices that can be integrated at or near the point of care, that is, at the time and place of patient care [49]. Microfluidic chips can function as standalone biosensing devices or can be integrated with other biosensing setups to make them more like POC devices [50]. These biosensors can be electrical, optical, electrical, electrochemical, magnetic, or colorimetric biosensors [50]. They can have many different types of microchannel depending on the application to which they will be put; these channels include laminar flow channels, hydrodynamic focusing channels, droplet generation channels, serpentine channels, spiral channels, gated parallel channels, mixing channels, centrifugal biochip channels, microstructured surface, and wells and grooves in channels [45,51,52,53]. All of these properties make microfluidic chips very diverse, dynamic, and adaptable systems for biosensing applications. 

Great efforts have been made to integrate microfluidics, microelectrochemical systems, nanotechnology, and materials science to facilitate the accuracy and economy of diagnostics. As a result, miniaturized, automatic, and integrated technologies have been developed and are becoming the desired substitutes for traditional methods to perform rapid, low-cost, accurate, and on-site (POC) diagnosis. Several such microfluidic platforms integrated with various techniques have emerged, such as lab-on-chip (LOC) devices.

#### 3.2.1. Lab-on-Chip (LOC)

Klostranec et al. developed a diagnostic system by combining quantum dots and microfluidics which resulted in a powerful tool for analyzing infectious agents in human serum samples in a multiplexed, high-throughput manner. This system has demonstrated its ability to detect serum biomarkers of widely prevalent bloodborne infectious diseases, such as hepatitis B, hepatitis C, and HIV, with remarkable sensitivity and speed. The system is designed to work with small sample volumes, requiring less than 100 µL of serum. This feature makes it a valuable tool for clinical settings where sample volumes may be limited. Additionally, the system’s rapidity allows for efficient analysis of large sample sets, making it an attractive option for screening purposes. One of the key advantages of this system is its high sensitivity, which is up to 50 times higher than that of other methods. This feature allows for the detection of low levels of infectious agents, which may not be detectable using traditional methods. The high sensitivity is achieved through the use of quantum dots, which are highly fluorescent nanoparticles that emit bright signals when bound to target biomolecules (Figure 3) [33]. This innovative device has the potential to be refined into a handy POC diagnostic tool, representing a significant advancement in the detection, monitoring, treatment, and prevention of infectious disease spread in developing countries. 

A microfluidics-based POC device to detect amplified nucleic acids from a bacterium related to tuberculosis was recently proposed by Liong et al. The device can perform DNA amplification, Magnetic NPs-DNA incubation, washing, and nuclear magnetic resonance (NMR) detection [54]. Although the NMR detection is fast, the amplification and incubation steps prolong the duration of the assay, but this technique is still faster than other tuberculosis detection methods based on cell culture and microscopy. In addition, a lab-on-chip platform has been created to detect Mycobacterium cells, utilizing a unique combination of an electric field, flow, and immunoaffinity binding. This novel concentration method allows for the swift detection of target bacteria in sputum samples, without the need for bacteriological culture, centrifugation, or nucleic acid amplification [55,56]. The platform’s minimal power requirement, which is only 5 W, and low cost make it ideal for point-of-care (POC) screening in settings with limited resources. This feature makes it possible to bring the test to the patient, reducing the need for expensive and time-consuming laboratory analysis. 

#### 3.2.2. Laboratory on a Cartridge Chip (LOCC)

Micrototal analysis systems, such as the LOCC technique, are widely used for high-throughput screening and multiple testing due to their fast analysis, low sample volume, and efficient manipulation capabilities. Various diagnostic protocols have been developed using microfluidic devices, such as Sun et al.’s smartphone-compatible nucleic acid amplification chip [57] and Kang et al.’s ultrafast on-chip PCR for SARS-Cov-2 detection [58]. Another approach based on surface-enhanced Raman spectroscopy has been proposed, which utilizes microchannels functionalized with aurum/argentum-coated carbon nanotubes or disposable electro-spray micro/nano-filter membranes to improve viral titer and accurately identify viruses from biological fluids/secretions [59].

### 3.3. Lateral Flow Immunoassay Chips

The most commonly used technique for rapid diagnostic testing is the lateral flow immunoassay (LFIA), due to its low cost, ease of use, and accessibility. The LFIA strip consists of several components, including the sample application pad, conjugate pad, nitrocellulose membrane, and adsorbent pad. When a sample is applied to the sample application pad, the biomolecule conjugate is released from the conjugate pad and moves through the nitrocellulose membrane, where it binds to the primary biomolecule against the analyte at the test line. The control line contains the secondary biomolecule, which binds to the released biorecognition molecule, and a color appears to indicate that the test is working correctly. This technique has been widely used for fast response in rapid diagnostic serological tests [60].

To improve the sensitivity and specificity of LFIA, Wang et al. introduced an amplification-free fluorescence assay using DNA probes and fluorescent nanoparticle-labeled monoclonal antibodies. This assay resulted in 100% sensitivity and 99.5% specificity in clinical trials for detecting the SARS-CoV-2 genome [61]. Similarly, Han et al. developed a dual-functional LFIA biosensor to detect the SARS-CoV-2 S1 protein. The biosensor contains SiO2 cores with 20 nm Au nanoparticles and quantum dots, which allow for detection via both the naked eye and fluorescence. The colorimetric function enables on-site diagnosis, while the fluorescence function can quantify the virus for critically ill patients (Figure 4), with extremely low detection limits for the S1 protein [62]. 

## 4. Non-Infectious Diseases

Non-infectious diseases encompass a range of health conditions that are not caused by pathogens, but rather by genetic, environmental, or behavioral factors. Examples of such diseases include cystic fibrosis, most cancers, cardiovascular diseases, chronic respiratory diseases (such as asthma), and diabetes mellitus [63]. These non-infectious diseases are responsible for 74% of all deaths worldwide, with 77% of these deaths occurring in low- and middle-income countries. Cardiovascular diseases are the leading cause of non-infectious disease deaths, followed by cancer, chronic respiratory diseases, and diabetes [64]. Early and accurate diagnosis of non-infectious chronic diseases is crucial for their effective management and improved clinical outcomes, thus promoting public health. 

### 4.1. Diabetes Mellitus

Diabetes mellitus (DM) is a collection of metabolic disorders characterized by insufficient insulin secretion, decreased insulin action, or a combination of both, leading to hyperglycemia [65]. The primary subtypes of DM are Type I and Type II, classified based on their pathogenesis involving insulin deficiency and/or resistance (Figure 5) [66]. Additional types of diabetes include gestational diabetes and neonatal diabetes. Rapid urbanization and lifestyle changes, along with unhealthy eating habits, are believed to contribute to the increasing prevalence of diabetes worldwide. If left untreated, diabetes can cause high blood sugar levels, which can damage blood vessels and nerves over time, resulting in various health complications such as nephropathy, retinopathy, neuropathy, cardiovascular disease, and strokes [66,67]. 

Diagnosis plays a critical role in managing diabetes as it enables the early identification of individuals who need interventions to prevent dysglycemia or complications. The diagnosis of diabetes traditionally relied on plasma glucose concentration and patient symptoms, with tests such as the fasting plasma glucose (FPG) and oral glucose tolerance test (OGTT) being established and validated [68]. However, Hemoglobin A1c (HbA1c) has emerged as a useful glycemic biomarker that correlates strongly with complications due to hyperglycemia. HbA1c testing is more convenient in clinical practice as it does not require fasting and can be carried out at any time, while also being a better predictor of long-term complications [66]. Nevertheless, HbA1c has limitations, such as its invasive blood sampling method, lower clinical sensitivity at the designated diagnostic threshold, and susceptibility to alterations by age, race, ethnicity, and clinical conditions. Moreover, its limited availability and cost make it impractical for routine use in some developing countries [66]. Given that there is no perfect biomarker for all diabetes patients in all situations, a more effective approach is to identify new biomarkers and use multivariate panels that combine different biomarkers to improve glycemic control, instead of seeking a single universal gold standard. 

### 4.2. Point-of-Care Testing Assays 

The focus of research related to diabetes diagnosis has centered on developing more efficient and effective methods due to the high impact of these diseases on society and the health sector. Biosensor technologies have made significant strides in recent years, leading to the creation of point-of-care (POC) diagnostics that are faster, simpler, affordable, and suitable for home measurements. The inclusion of several nanosystems into detection tools has further improved POCTs, resulting in sensitive, cost-effective, and user-friendly detection schemes. Nanotechnology has also enabled the development of advanced and superior POC diagnostic devices for diabetes that allow for non-invasive testing of several biomarkers, in contrast to conventional glucose monitoring methods [67,69]. 

Traditionally, blood glucose level assessments have been used for diagnosis, but this method has several disadvantages, such as its invasiveness, which is associated with pain and discomfort, leading to incompliance. Recently, exhaled breath acetone has emerged as a promising biomarker for diabetes due to its strong positive correlation with blood glucose levels. Traditional methods of detecting acetone, such as gas chromatography mass spectrometry (GC-MS), proton transfer reaction mass (PTR-MS), and other laser-based techniques, are highly sensitive and selective. However, they rely on sophisticated equipment, complex sample collection methods, and are only available in advanced medical facilities, making them expensive and impractical for on-site testing. To address these limitations, researchers have turned to biosensors for breath acetone detection over the past decade [67]. 

### 4.3. Nanomaterials 

Transition metal ions possess unfilled d-shells, allowing for reactive electronic transitions, wide band gaps, superior electrical characteristics, and high dielectric constants. As a result, metal oxide nanomaterials (MONs) have exceptional and adjustable optoelectronic, optical, electrical, thermal, magnetic, catalytic, mechanical, and photochemical properties. As sensing materials, MONs, mainly copper(II) oxide (CuO), copper(I) oxide (Cu2O), tin(II) oxide (SnO), tin(IV) oxide (SnO2), zinc oxide (ZnO), nickel oxide (NiO), indium oxide (In2O3), and tungsten oxide (WO3), are promising candidates that manifest high sensitivity, fast response/recovery (res/rec) time, excellent reproducibility and stability, and cost-effectiveness with simple fabrication processes [70]. 

Electrochemical, optical, and mass-based biosensors have been developed for acetone detection in the breath, with the ideal breath acetone biosensor needing to detect acetone at the subpart per million level. For diabetes, the concentration of exhaled acetone is typically above 1.8 ppm, while healthy individuals have levels between 0.3 and 0.9 ppm. Chemiresistive gas sensors using semiconductor metal oxides, such as In_2_O_3_, CeO_2_, WO_3_, SnO_2_, and ZnO, have been widely used for their sensing capabilities. These sensors detect changes in electrical conductivity in the presence of gases and oxygen, which lead to catalytic reduction/oxidation reactions at the metal oxide surface. Chemiresistive gas sensors offer several advantages, including portability, suitability for small-scale applications, low power consumption, and low-cost mass production. They must also be able to function in high relative humidity environments since human exhaled breath typically contains around 80-90% relative humidity. To be user-friendly, they should operate at low temperatures, be environmentally stable, and use friendly sensing materials. To enhance the performance of these biosensors, nanotechnology has been integrated into their design [69].

### 4.4. Wearable Devices 

Wearable sensors have recently been introduced for non-invasive, rapid, and real-time monitoring of glucose levels in human body fluids [71,72,73]. These sensors generally utilize optical and electrochemical methods. The first electrochemical wearable glucose sensor was developed by Cygnus in 2001, known as the GlucoWatch, but it was not long-lasting, resulting in high costs and inconvenience for users. To address this, nanomaterials have been integrated into wearable glucose-sensing devices. Emaminejad et al. used carbon nanotubes as the immobilization matrix for GOD and H_2_O_2_ detection material, resulting in a perspiration-based wearable glucose sensor. The glucose was converted into gluconic acid and H_2_O_2_ by GOD in the presence of oxygen, and the glucose concentration could be extrapolated indirectly by detecting H_2_O_2_ [74]. On the other hand, Toi et al. created a wearable sensor with high glucose sensitivity using a composite fiber rGO/polyurethane (PU) modified with oxygen-containing functional groups. The sensor had a low detection limit of 500 nmol/L, high selectivity against interference, and high mechanical durability, as shown in Figure 6 [75].

Metal oxide nanomaterials (MONs) are attractive in the construction of flexible/wearable sensors due to their tunable band gap, low cost, large specific area, and ease of manufacturing [30,70]. Recently, In_2_O_3_-nanomaterial-based flexible glucose sensors employing the enzymatic oxidation of d-glucose with glucose oxidase have been reported, where the concentration of d-glucose was estimated through the pH level determined by the concentration of hydrogen ions generated via enzymatic glucose oxidation. The real-time glucose monitoring technology can be developed for generalized healthcare applications, including preliminary medical diagnosis and chronic self-management. Moreover, a wearable In2O3 nanoribbon transistor biosensor was developed, in which chitosan, carbon nanotubes (CNTs), and glucose oxidase were coated on the source and gated electrode of an In2O3 FET to detect glucose in body fluids [76]. Jiang et al. studied Cu_2_O-based wearable glucose sensors with various morphologies and reported that the cuboctahedral Cu_2_O-based sensor exhibits the highest selectivity and responsive signal. This finding highlighted that the shape of nanoparticles affects their properties in wearable sensors due to the different facets. Consequently, the team developed a wireless real-time glucose monitoring system based on a wearable Cu_2_O NP sensor which transmitted the concentration data to a smartphone via Bluetooth connection [77]. 

The development of biosensors capable of detecting glucose in other biofluids such as tears and urine is an exciting area of research, as it could revolutionize point-of-care glucose monitoring for individuals with diabetes. The use of glucose-responsive photonic crystal hydrogels shows promise as a simple and cost-effective optical sensor for detecting glucose levels. A polyelectrolyte photonic hydrogel with GOx immobilization has been investigated as a naked-eye optical sensor. The cationic polyelectrolyte poly(2-dimethylaminoethyl methacrylate) expands and has a clear pH response in the form of a red shift with decreasing pH, while the enzymatic reaction of GOx with glucose in the hydrogel results in a reduction in pH. Furthermore, Elsherif et al. introduced phenylboronic acid (PBA) groups into a photonic microstructural hydrogel and used it to fabricate a wearable contact lens optical sensor in which the photonic crystal hydrogel expands when bound to glucose in tears. The integration of this technology into wearable contact lenses for continuous glucose monitoring is a particularly exciting development, as it offers a non-invasive and convenient way to monitor glucose levels in tears. This could provide individuals with diabetes with real-time glucose monitoring and alerts, potentially improving their quality of life and reducing the risk of complications associated with poorly managed diabetes. Overall, the development of glucose-responsive photonic hydrogels is an exciting area of research with promising applications in point-of-care glucose monitoring and diabetes management [78]. Table 2 presents a summary of selected naomaterial-based detection methods or devices for diagnosing diseases that are common in developing countries.

## 5. Emerging Trends and Future Outlooks

Various novel detection methods can be utilized to aid in the development of point-of-care (POC) diagnostics, including electrochemical biosensors, fluorescence biosensors, surface-enhanced Raman scattering (SERS)-based biosensors, colorimetric biosensors, surface plasmon resonance (SPR)-based biosensors, and magnetic biosensors [95]. These methods have been empowered by the booming development of microfluidics, microelectromechanical systems (MEMS) technology, and nanotechnology, which have substantially promoted the development of POC devices such as lab-on-a-chip (LOC), lab-on-a-disc (LOAD), lateral flow, miniaturized polymerase chain reaction (PCR), isothermal nucleic acid amplification (INAA), and microfluidic paper-based analytical devices (µPADs) [96]. The combination of novel biosensors, nanotechnology, and artificial intelligence (AI) techniques has the potential to accelerate the development of POC diagnostics toward more efficient and intelligent diagnosis. By seamlessly combining these technologies, it may be possible to automate diagnostic processes, reduce human error, and increase the accuracy and speed of diagnosis. Furthermore, the development of POC diagnostics using these technologies could potentially enable healthcare professionals to make quicker and more informed decisions, leading to improved patient outcomes and reduced healthcare costs. 

The cost/benefit analysis of the emerging trends and future outlooks discussed in this article indicates several advantages and challenges. One of the benefits of the emerging trends in POC diagnostics is the potential for more efficient and intelligent diagnosis, which can lead to quicker and more informed decisions by healthcare professionals. This can result in improved patient outcomes and reduced healthcare costs. Additionally, the combination of novel biosensors, nanotechnology, and AI techniques can automate diagnostic processes, reduce human error, and increase the accuracy and speed of diagnosis. AI algorithms, such as machine learning and deep learning, can be applied to medical data analysis to generate insights that can improve patient outcomes and reduce healthcare costs [97]. AI can aid in medical diagnosis, drug development, and patient care, and can revolutionize the way medical data are analyzed, leading to faster and more accurate diagnoses. 

Bioengineers working in the field have recognized the potential impact of nanotechnology and AI on global health, leading to the development of numerous strategies for pathogen tracking and monitoring, drug delivery, and disease prevention. Despite significant advancements in diagnostic tools in the past decade, certain recognition materials continue to face challenges such as cross-reactivity and high production costs. Biosensors commonly employ nanomaterials with intricate structures to enhance detection. However, these nanomaterials often exhibit varying morphologies between batches, leading to poor reproducibility. Therefore, further research is necessary to gain a better understanding of the nanomanufacturing process, control particle aggregation, and regulate surface interactions. Strict quality control measures are also essential for clinical applications to guarantee reliable and accurate measurements. 

Another challenge in biological analysis is detecting multiple molecules at once, as current assays are often limited to a single analyte. However, studying multiple substances simultaneously can provide more comprehensive information and enhance detection efficiency, especially in cases where sample volume is scarce or invasive to collect. To address this, multiplex assays using nanomaterials of varying size and shape, or fluorescent nanoparticles with different emission wavelengths, can be employed to detect multiple biomarkers in a single measurement. By doing so, researchers can better understand how different molecules work together to regulate biological processes. Future directions for AI in the field include improving the accuracy and speed of medical diagnosis, the further development of drug discovery, and the integration of AI with POC devices for more efficient and effective healthcare delivery.

Lastly, in developing countries, the problem of accessibility to healthcare is greatly exacerbated, where large populations have little or no access to even the basic health services due to financial limitations, a shortage of skilled personnel, and a lack of infrastructure. The lack of infrastructure is not only with respect to biomedical and clinical equipment, but may also include access to running water, refrigeration, and electricity. In addition to basic medical tests and screening for chronic disease, affordable test kits for infectious diseases can be a life-saving intervention in many developing countries, where millions die every year due to inadequate diagnosis and these tests could help prevent epidemics from turning into pandemics. For this reason, the use of cost-effective material in terms of affordability and durability is highly recommended for biosensors intended for developing countries. Strategies might include the following: the development of inexpensive POC testing devices operated with solar-rechargeable batteries instead of electricity; the creation of more devices manufactured with abundant, locally available raw material, e.g., paper-based LFAs; and the introduction of POC testing devices with assays that require minimal or no sample preparation and the use of smartphone cameras as spectrographs instead of expensive spectrometers for obtaining results. All these new directives could be of great use in improving the healthcare system and improving the disease survival rate in developing countries.

## 6. Conclusions

This review discussed recently developed POC devices that have been improved with nanomaterials to improve their function for the effective, convenient, and affordable detection of diseases in low income/low resources countries. With nanotechnology, diagnostics are performed at the nanoscale, resulting in the use of handheld devices that are easy to operate and affordable. Nanoparticle-based platforms have been developed and optimized for the detection of infectious and non-infectious disease biomarkers, making diagnostic procedures less cumbersome with elevated sensitivity because most complex procedures are now integrated onto a simple device having the capacity to be used for on-the-spot disease detection even in low-resource areas. Multiplexing is currently sought-after, which is attainable through the integration of nanomaterials and artificial intelligence. However, the development of cost-effective POC devices applicable in developing countries is on the rise but remains constrained by challenges. 

## Figures and Tables

**Figure 1 nanomaterials-13-01247-f001:**
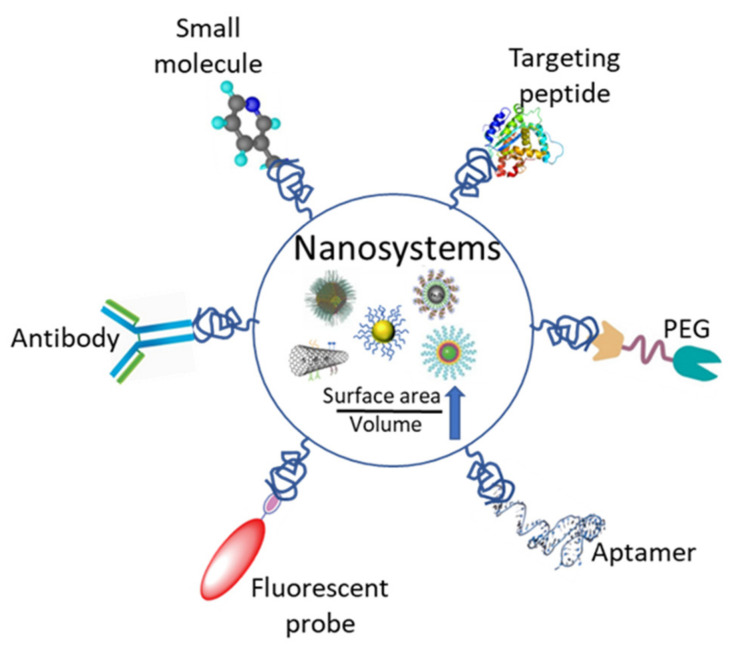
A representation of the various functionalization moieties for nanosystems utilized for the detection of biomolecules in diagnostics.

**Figure 2 nanomaterials-13-01247-f002:**
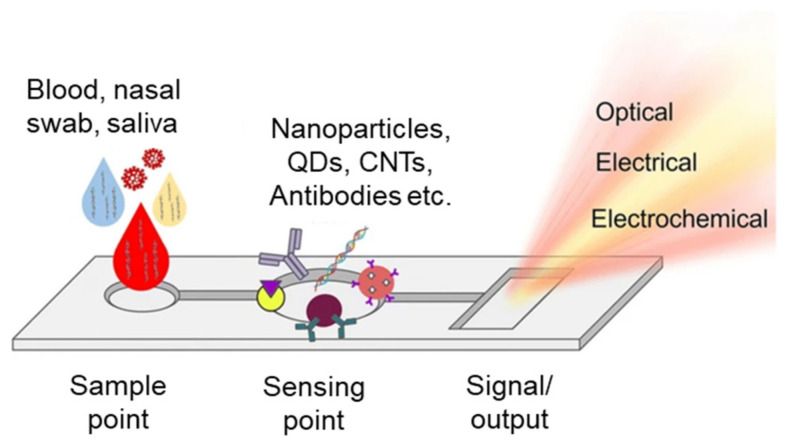
Basic representation of a microfluidic chip depicting the sample receiving point, the sensing point, and the signal/output point.

**Figure 3 nanomaterials-13-01247-f003:**
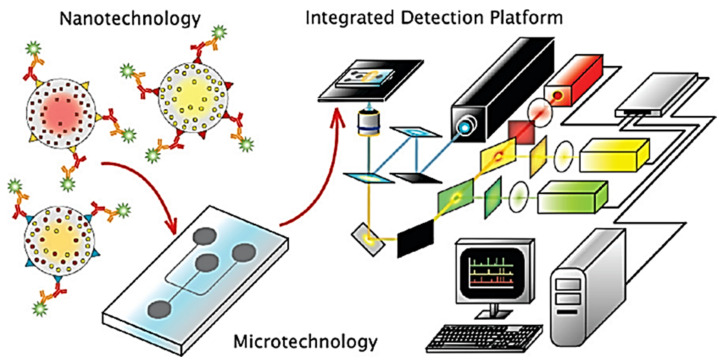
Illustration of the integration of quantum dots, solution-based sandwich assay, microfluidics and fluorescence detection with custom software for high-throughput, multiplexed blood-borne pathogen detection. Reproduced with consent from [33]. Copyright 2007 American Chemical Society.

**Figure 4 nanomaterials-13-01247-f004:**
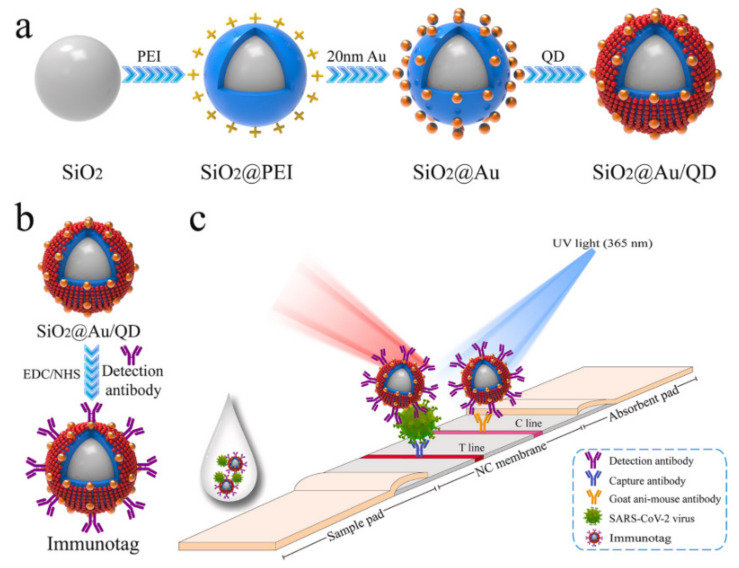
A schematic representation of a fabrication process for dual-functional SiO_2_@Au/QD fluorescent labels (**a**,**b**) with colorimetric and fluorescent functionality to detect the SARS-CoV-2 S1 protein by dual modes of naked eye / fluorescence (**c**). Reproduced with consent from [62].

**Figure 5 nanomaterials-13-01247-f005:**
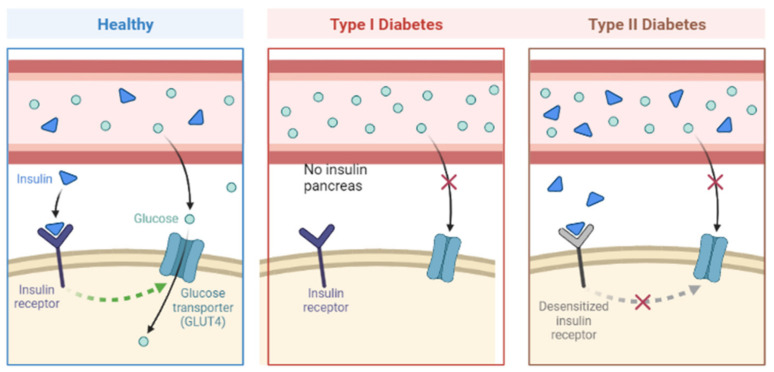
Schematic representation of diabetes mellitus Type I and Type II pathogenesis compared to a healthy condition. Drawn using BioRender.

**Figure 6 nanomaterials-13-01247-f006:**
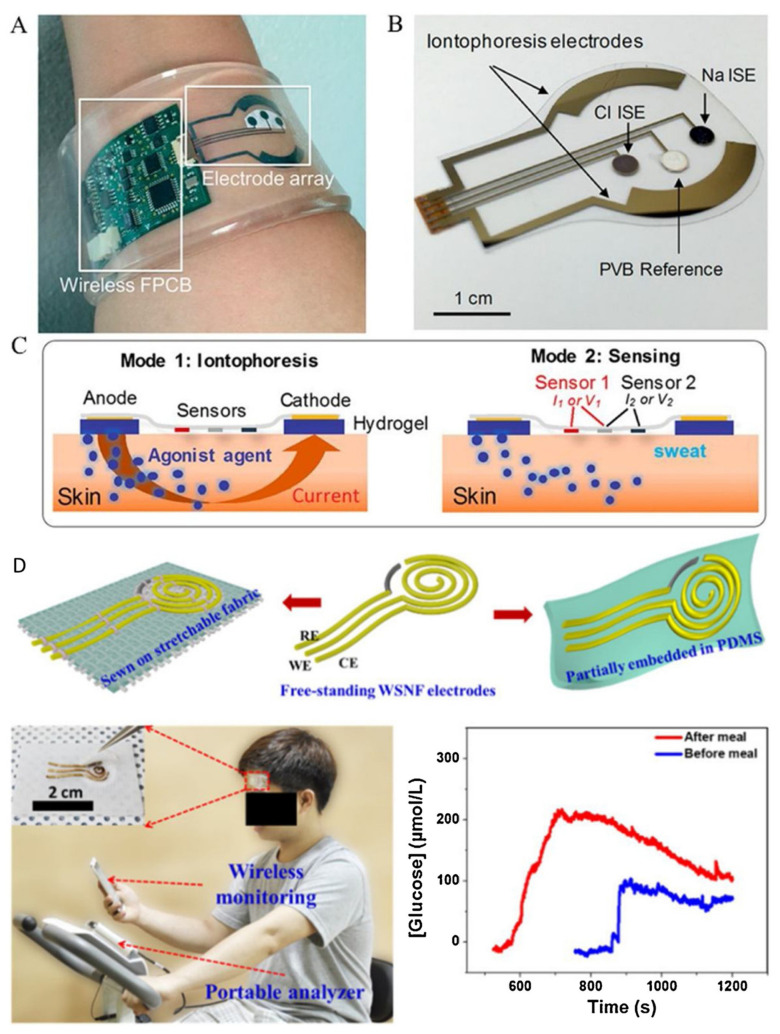
(**A**–**D**) Examples of perspiration-based wearable glucose sensors. Reproduced from Ref [73] with permission from Elsevier.

**Table 1 nanomaterials-13-01247-t001:** Nanomaterials and their common applications in diagnostics.

Nanomaterial	Description	Potential Applications
**Quantum dots**	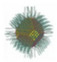 Quantum dots are semi-conductor nanocrystals that possess excellent optical and electrical properties. Bright and photostable for the detection and detection of biomarkers.	Optical detection of proteins Tumor visualization
**Carbon nanotubes**	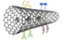 Carbon nanotubes are made of a layer of graphene that is rolled into a cylinder. Functionalized/decorated with different moieties for biomedical applications.	DNA mutations detectors Disease protein biomarker detection
**Nanoparticles**	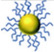 Metallic, polymeric, and layered double hydroxide nanoparticles. Properties tailored by size, shape, composition, and surface modification.	Disease protein biomarker detection Detection of DNA mutations Targeted drug delivery Imaging agents
**Nanoshells**	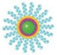 Spherical nanoparticles made of a di-electric core that is capped by a thin metallic shell (gold). Nanoshells have a quasiparticle known as a plasmon.	Tumor Imaging or Visualization

**Table 2 nanomaterials-13-01247-t002:** A summary of selected nanomaterial-based detection methods for diseases prevalent in developing countries.

Type of Nanomaterial	Detection Analyte	Detection Methodology	Targeted Disease	Reference
Gold nanoparticles	Mycobacterium tuberculosis	Electrochemical	Tuberculosis	[79]
Silica nanoparticles	HIV-1 p24 antigen	Photochemical	HIV/AIDS	[80]
Magnetic nanoparticles	Plasmodium falciparum	Electrochemical	Malaria	[81]
Gold nanoparticles	SARS-CoV-2/N gene RNA	Colorimetric	COVID-19	[82]
Carbon nanotubes	Glucose	Amperometric	Diabetes	[83]
Graphene oxide	Prostate-specific antigen	Fluorescence	Prostate cancer	[84]
Magnetic nanoparticles	DNA	Electrochemical	Genetic disorders	[85,86]
Quantum dots	Carcinoembryonic antigen	Optical	Colon cancer	[87]
Gold nanoparticles	SARS-CoV-2 pseudo virus	Colorimetric	COVID-19	[88]
Magnetic nanoparticle	*E. coli*	Microscopy	Foodborne disease by *E. coli*	[89]
Silver nanoclusters	S. typhimurium	Fluorescence	Typhoid	[90]
Silver nanoclusters	*E. coli*	Fluorescence	Bacterial infection	[91]
Gold nanoparticles	Listeria monocytogenes	Fluorescence	listeriosis	[92]
Gold nanoparticles	Proteus mirabilis in urine	Colorimetric	Urease producing bacteria	[93]
Gold nanoparticles	IL-6 in blood	Colorimetric	Sepsis	[94]

## Data Availability

Data are available via personal communication with proper reasons.

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
