# Peer review of "Nanotechnology-Based Diagnostics for Diseases Prevalent in Developing Countries: Current Advances in Point-of-Care Tests"

_nanomaterials, 2023, doi:10.3390/nano13071247_

Round 1
Reviewer 1 Report
The article entitled “Nanotechnology-based diagnostics for diseases prevalent in developing countries: current advances in point-of care tests” aims to review recent the developments in POCT devices that benefit nanotechnology. ” But the “addresses the challenges faced by these technological advances and interesting future trends” topic is weak. The authors may present a perspective of cost/benefit of each. Which is the sensibility? Specificity for analyte? Can be used low or high tech?
Author Response
Response uploaded in pdf file.

Reviewer 2 Report
The review article by Thwala et al. provides a brief study on the recent developments of point-of-care (PoC) devices based on different nanomaterials for the early diagnosis of diseases that are predominant in developing countries. The topic of research is interesting and appealing. However, the manuscript needs some major amendments as follows-
Authors are suggested to summarize important papers relevant to the research topic in Table(s) highlighting useful parameters such as type of nanomaterial used, detection analyte, detection methodology, targeted disease, etc. for the convenient of readers.
In recent years, several review papers have been published on the similar topic (Biosensors 2022, 12, 737, Sensors 2022, 22(4), 1620, Biosens. Bioelectron. 2019, 126, 68, etc.). Authors need to discuss them in the Introduction Section and emphasize what new knowledge is gain from the current review paper. What does it add to the subject area compared with other published material? What is the main question addressed by this research? Moreover, authors also need do proper literature survey and cite the missing recent important review papers and articles related to the topic (For e.g., Chem. Soc. Rev. 2018, 47, 4697, Biosens. Bioelectron. 2017, 87, 373, Adv. Mater. Technol. 2019, 4(9), 1900361, Front. Chem. 2020, 8, 586702, Anal. Chem. 2022, 94, 10685).
Nanoparticle section needs major improvements which is crucial considering the scope of journal. Note that magnetic nanoparticles also come under the category of metallic nanoparticles. Authors need to elaborate this section more by discussing other important emerging nanoparticle systems such as layered double hydroxides (LDH) nanoparticles, conjugated polymer nanoparticles (CPNs), metal oxide nanoparticles, (Nanomaterials 2019, 9(10),1404, Microchim. Acta 2022, 189, 83, ACS Nanosci. Au 2022, 2, 2, 64) which are also promising for diagnostic applications.
On Page 7, Line 288: Write “Klostranec, JM” as Klostranec et al.
The formatting of References is improper need attention.
Author Response
Responses uploaded in pdf file.

Round 2
Reviewer 1 Report
The authors have performed appropriated modifications.
Reviewer 2 Report
The authors have satisfactorily answered most of my previous comments. There are only some minor corrections such as SnO2, Cu2O, WO3 etc. should be written as SnO2, Cu2O, WO3, which I believe can be handled during the proofreading. I appreciate their efforts and recommend the current version of manuscript for publication.